# Sambucus Nigra Extracts–Natural Antioxidants and Antimicrobial Compounds

**DOI:** 10.3390/molecules26102910

**Published:** 2021-05-14

**Authors:** Anna Przybylska-Balcerek, Tomasz Szablewski, Lidia Szwajkowska-Michałek, Dariusz Świerk, Renata Cegielska-Radziejewska, Zbigniew Krejpcio, Elżbieta Suchowilska, Łukasz Tomczyk, Kinga Stuper-Szablewska

**Affiliations:** 1Department of Chemistry, Poznań University of Life Sciences, 60-637 Poznan, Poland; anna.przybylska@up.poznan.pl (A.P.-B.); kinga.stuper@up.poznan.pl (K.S.-S.); 2Department of Menegment of Food Quality and Safety, Poznań University of Life Sciences, 60-637 Poznan, Poland; tomasz.szablewski@up.poznan.pl (T.S.); renata.cegielska-radziejewska@up.poznan.pl (R.C.-R.); lukasz.tomczyk@up.poznan.pl (Ł.T.); 3Department of Landscape Architecture, Poznań University of Life Sciences, 60-637 Poznan, Poland; dariusz.swierk@up.poznan.pl; 4Department of Human Nutrition and Dietetic, Poznań University of Life Sciences, 60-637 Poznan, Poland; zbigniew.krejpcio@up.poznan.pl; 5Department of Genetics, Plant Breeding and Bioresource Engineering, University of Warmia and Mazury in Olsztyn, 10-719 Olsztyn, Poland; ela.suchowilska@uwm.edu.pl

**Keywords:** *Sambuci fructus*, double hydrolysis, the antibacterial activity of the elderberry fruit extracts

## Abstract

Due to the health-promoting properties of elderberry fruits, which result from their rich chemical composition, this raw material is widely used in herbal medicine and the food industry. The aim of the study was to demonstrate the antibacterial activity of the elderberry fruit extracts. The research showed that the content of phenolic acids and flavonoids in the extracts determined their antibacterial activity. The research showed that the content of phenolic acids and flavonoids in the extracts determined their antibacterial activity. The following phenolic acids were predominant: chlorogenic acid, sinapic acid, and t-cinnamic acid. Their average content was, respectively, 139.09, 72.84, 51.29 mg/g extract. Rutin and quercetin (their average content was 1105.39 and 306.6 mg/g extract, respectively) were the dominant flavonoids. The research showed that the elderberry polyphenol extracts exhibited activity against selected strains of bacteria within the concentration range of 0.5–0.05%. The following bacteria were the most sensitive to the extracts: *Micrococcus luteus*, *Proteus mirabilis*, *Pseudomonas fragii*, and *Escherichia coli*. Of the compounds under analysis, apigenin, kaempferol and ferulic, protocatechuic, and p-coumarin acids had the greatest influence on the high antibacterial activity of elderberry extracts. The results of the microbiological and chemical analyses of the composition of the extracts were analyzed statistically to indicate the bioactive compounds of the greatest antimicrobial significance.

## 1. Introduction

Elderberry is a plant species of the *Adoxaceae* family. Formerly it was also included in the *Sambucaceae* and *Caprifoliaceae* families [1,2]. Currently there are about 20 intraspecific taxa of elderberry around the world. The varieties growing in forests and parks are predominant, but there are also cultivars grown for industrial purposes [3,4,5,6,7].

Elderberry fruits (*Sambuci fructus*) are spherical drupes, occurring in the form of corymbs with several dozen pieces inside. Their color depends on the stage of ripeness. In the beginning the fruit is green, but when it is fully ripe, it is black and violet [7,8,9]. Due to the health-promoting properties of elderberry fruits, which result from their rich chemical composition, this raw material is widely used in herbal medicine and the food industry.

The chemical composition of elderberry fruits depends on various factors, including the cultivar, environmental conditions (light, temperature, amount, and frequency of rainfall, fertilization, cultivation methods), the processing method and storage conditions. Ripe elderberry fruits contain carbohydrates, including dietary fiber, mainly hemicelluloses and polygalacturonic acid, pectins, and simple sugars [10,11]. Elderberries contain small amounts (2.7–2.9%) of protein, including seven exogenous amino acids [12,13]. They are also rich in fatty acids, mainly linoleic acid, linolenic acid, and oleic acid [11,14]. Elderberry fruits are a source of tannins and organic acids, including malic, acetic, valeric, tartaric, shikimic, and benzoic acids [15,16]. Apart from that, elderberry fruits contain 0.01% of essential oil, which is composed of over 30 different compounds, including phenyl aldehydes (3–25.8% of the oil composition) and furfural (18%) [17,18]. Elderberry fruits are also a source of minerals [19,20,21,22], considerably diversified amounts of vitamin C: 6–25 mg/100 g [23], 18–36 mg/100 g [21], 18–26 mg/100 g [24], 6–44 mg/100 g [19,20], and even 132.1 mg/100 g [17], as well as a wide range of B vitamins: B_2_–65 mg/100 g; B_7_–17 mg/100 g, B_9_–1.8 mg/100 g [21,24] and vitamin A (600 IU/100 g) [21].

The antioxidative properties of elderberries are mostly determined by anthocyanins, which are a large group of bioactive compounds (cyanidin-3-glucoside: 65.7% of all anthocyanins) [25,26], as well as flavonoids, flavonols, and phenolic acids.

According to reference publications, there are significant differences between the profile of bioactive compounds in the fruits of cultivated and wild plants [27]. Currently Poland is a country with the best-documented tradition of picking wild edible plants [28,29]. The progressive nutritional trends oriented to traditional functional food have sparked intensified research on wild edible plants [30,31]. The health-promoting and healing properties of elderberry products and preparations were the basis of this study. So far, the antimicrobial activity of ethanol-aqueous extracts from freeze-dried elderberry concentrates has been investigated on 13 nosocomial pathogens, including *Staphylococcus aurens*, *Bacillus cereus*, *Salmonella poona*, and *Pseudomonas aeruginosa* [32]. However, currently the most important direction of research is the compounds contained in elderberry fruit products, which exhibit antiviral activity, especially against group A (KAN-1 HSNI) and B viruses (B/Mass) [33,34].

Literature data reported that extracts of black elder flowers as well fruits inhibited Gram-positive (*S. aureus*, *B. cereus*) and Gram-negative bacteria (*S. poona*, *P. aeruginosa*). Flavonons, flavonols dihydroflavonols presentin fruts, and flowers of black elder may be responsible for their antimicrobial properties. Furthermore, they can contain lecithin, peptides, and oligosaccharides which are inhibitors of transcription and metabolism of the bacterial cells [32].

This study was conducted on extracts of bioactive compounds from freeze-dried elderberry fruits growing wild in Poland. The extracts were obtained with an original method described in detail in the Materials and Methods section.

The aim of this study was to assess the antimicrobial activity of aqueous solutions of wild elderberry extracts obtained by double hydrolysis. The results of the microbiological and chemical analyses of the composition of the extracts were analyzed statistically to indicate the bioactive compounds of the greatest antimicrobial significance. Interdisciplinary research was conducted to find interrelations between the factors under analysis and to better understand the antimicrobial activity of elderberry extracts.

## 2. Results and Discussion

The samples of elderberries collected from 36 locations in Poland were subjected to the extraction process described in the Materials and Methods section. The extract preparation method was optimized to match the specificity and chemical composition of the material. The main focus of the investigations was bioactive compounds, mainly antioxidants. The average values of the results are shown below.

The first stage of the investigations involved analysis of the antioxidative activity measured by means of ABTS^+^ (Table 1). A high value of the antioxidative activity indicated the presence of antioxidative compounds. The next stage of investigations involved measurement of the total free phenolic acids content and total flavonoids content (TFC). The elderberry extracts had a high amount of these compounds. By comparison, according to reference publications, the content of polyphenols per 100 g of elderberry fruits is diversified, e.g., 25.87–38.87 mg/g DM [23], 827 mg [35], 1336 mg [36], or 513.6 mg [37].

The next stage of the investigations involved analysis of the content of 15 selected phenolic acids (Table 2) and 8 flavonoids (Table 2) in the extracts (Figure 1). The following phenolic acids were predominant: chlorogenic acid, sinapic acid, and t-cinnamic acid. Rutin and quercetin were the dominant flavonoids. However, it is noteworthy that the antimicrobial activity of these extracts results from the entire composition of bioactive compounds rather than from individual compounds, as has been presented in scientific publications so far. According to the reference publications, polyphenols contained total flavonoids at a concentration of 186 mg/100 g in, where quercetin (about 73.4 mg/100 g) [14], isoquercitin, rutoside, and hyperoside were the dominant compounds [11,38]. Another two groups of bioactive compounds described in the reference publications were total flavonols (38.26–142.3 mg/100 g) [39,40] and total phenolic acids (20.00 mg/100 g) [37]. Like in our research, chlorogenic acid was also the dominant phenolic acid in other studies (10–32 mg/100 g) [37,38,41].

Organic acids were the next group of compounds analyzed in our research. Table 3 lists the content of organic acids and basic sugars: glucose and fructose. These parameters indicate the degree of fruit ripeness and determine the organoleptic characteristics of fruit products. The relation between acids and sugars should be inversely proportional. On the one hand, a higher sugar content makes fruits and fruit products more desirable. On the other hand, a high content of organic acids reduces the pH and stabilizes the bonds and structures of phenolic acids, flavonoids, and anthocyanins. In consequence, the antioxidative activity of the entire system of bioactive compounds increases. Citric acid was the dominant acid, which is characteristic of berries.

At the next stage the elderberry extracts were analyzed for the content of natural pigments determining the color of ripe fruits. The analysis revealed the presence of carotenoids, chlorophylls, and anthocyanins (Table 4). Anthocyanins were the compounds with the highest total mean content, i.e., 109 mg/g of the elderberry fruit extract. According to the data provided in the reference publications, the total anthocyanins content in 100 g of fresh elderberries was considerably diversified, i.e., 200–1000 mg [42], 465.1 mg [37], 272.87 mg [40], 863.89 mg [43], 1265 mg [20]. The differences in the content of anthocyanins may have been caused by differences between the varieties of elderberry fruit and the weather conditions during ripening [44]. The authors of other studies mostly identified: cyanidin-3-glycoside (65.7% of all anthocyanins), cyanidin-3-sambubioside (32.4%), and cyanidin-3-diglycoside (0.8% of all anthocyanins) [26,45]. According to the data provided in the reference publications, the content of anthocyanins in 100 g of fresh fruit amounted to 529–664 mg in the Haschberg cultivar, 877 and 1815 mg in the Sampo cultivar, and 846 and 1634 mg in the Samyl cultivar. The content of anthocyanins in elderberry products depended on the degree of processing and technological conditions, since anthocyanins are unstable compounds [11,44,46]. Elderberries usually contain 0.2–1% of anthocyanins and about 0.01% of essential oil, which contains about 50 compounds. For this reason, elderberries are used as a natural pigment and for the flavoring of food products [13,34,43]. Additionally, unripe fruits contain sambucin, and more precisely–sambunigrin. This is a highly poisonous cyanogenic glycoside (benzaldehyde cyanohydrin). Therefore, the consumption of raw elderberries may cause acute poisoning, weakness, and gastrointestinal ailments [21]. The results of toxicological studies led to the conclusion that the consumption of elderberries after thermal treatment is safe and does not cause any side effects [47,48,49,50].

The composition of extracts of phenolic compounds is an important determinant affecting various directions of their activity. This study focused on the antibacterial effect of the extracts.

Reference strains of the bacteria which are pathogenic mainly to humans and cause food spoilage were selected for microbiological tests (Table 5). Before some of these bacterial strains were not investigated thoroughly enough for their sensitivity to the activity of polyphenolic bioactive compounds.

The research showed that the elderberry polyphenol extracts exhibited activity against selected strains of bacteria within the concentration range of 0.5–0.05%. The following bacteria were the most sensitive to the extracts: *M. luteus*, *P. mirabilis*, *P. fragii*, and *E. coli*. This finding significantly broadened the current knowledge, because so far only selected tannins derived from hydroxycinnamic, gallic, and caffeic acids, as well as triterpenes (oleanic acid and α- and β-amarin) were considered antimicrobial substances [14,51]. The studies conducted so far showed that selected polyphenols and their esters inhibited the growth of bacteria of the following genera: *Yersinia*, *Bacillus*, *Corynebacterium*, *Proteus*, *Staphylococcus*, *Enterococcus*, *Klebsiella*, *Micrococcus*, *Escherichia*, and *Pseudomonas*. Gallic, vanillic, synaptic, and protocatechuic acids effectively inhibit the growth of Gram-positive and Gram-negative bacteria, e.g., *E. coli*, *Enterobacter cloacae* DG-6, and *Ps. acidovorans* [52,53,54,55,56,57]. These acids are more effective against Gram-positive than Gram-negative bacterial cells, because the cells of Gram-negative bacteria have an outer shell surrounding the cell wall, which makes it difficult for hydrophobic compounds to diffuse through the liposaccharide membrane into the cell. Caffeic, ferulic, and protocatechic acids are bioactive compounds which inhibit the growth of bacteria responsible for food poisoning, e.g., *Bacillus subtilis* and *Bacillus cereus*. Phenolic acid derivatives also exhibit the bactericidal effect against *Y. enterocolitica* rods. Among them o-coumaric acid compounds are more effective than m-coumaric acid derivatives, which could be associated with both the chemical structure of phenolic compounds and the resistance of these bacteria [57]. Martins et al. [58] observed that polyphenol extracts inhibited the development of *Candida* species [59,60,61]. Kędzia and Hołderna-Kędzia [62] used phenolic acids, i.e., p-coumaric acid, caffeic acid, and ferulic acid in their research and observed that they inhibited the growth of *St. aureus*, *E. coli*, *K. pneumoniae*, *E. faecalis*, and *Ps. aeruginosa* bacteria. Salicylic acid exhibited a relatively stronger antibacterial effect (MIC 100–500 μg/mL). Gallic acid also exhibited a strong antibacterial effect (the MIC for *S. aureus* was 150 μg/mL). On the other hand, syringic, gentisic, and p-hydroxybenzoic acids had a very weak effect on *S. aureus* (MIC > 1000 μg/mL). The same authors also found that depsides (chlorogenic acid, rosmarinic acid, ellagic acid) exhibited weak antimicrobial activity against *S. aureus*, *E. coli*, *K. pneumoniae*, *E. faecalis*, and *Ps. aeruginosa* [62]. Efenberger-Szmechtyk et al. [63] also used aqueous extracts of phenolic acids to investigate their antimicrobial properties and observed that they effectively limited the growth of the following bacteria: *E. faecalis*, *B. thermosphacta*, *E. coli*, *Ps. fluorescens*, *L. rhamnosus*, *P. mirabilis*, and *E. aerogenes* [63]. Latest studies also confirmed the antibacterial activity of six phenolic acids and showed the antibacterial potential of extracts against Gram-positive bacteria (*E. faecalis* and *L. monocytogenes*) [64,65]. Camargo et al. [66] also noted that phenolic acid extracts inhibited the growth of Gram-positive bacteria (*B. cereus*, *S. aureus*, *L. monocytogenes*, *Gb. stearothermophilus*) and Gram-negative bacteria (*Ps. aeruginosa*, *Ps. fluorescens*, *S. enteritidis*, *S. typhimurium*, *E. coli*) [66]. Other researchers studied the activity of flavonoid extracts against Gram-negative bacteria, i.e., *E. coli* and *Ps. aeruginosa*, and found that the level of their activity depended on their chemical structure [67]. Malterund et al. [68] studied the antibacterial properties of flavonoids and found that only naringenin exhibited antibacterial activity against *E. coli*, *S. aureus*, and *E. faecalis*. Iwagawa et al. [69] noted that some quercetin derivatives inhibited the growth of *E. coli*. Basile et al. (1999) [70] found that apigenin, vitexin, and saponarin acted selectively against some Gram-negative bacteria, i.e., *P. vulgaris*, *P. mirabilis*, *Ps. aeruginosa*, *E. coli*, *K. pneumoniae*, and *E. cloacae*. The minimum inhibitory concentration of apigenin was determined in that study. The other flavonoids were not active against the strains tested. The same scientists conducted an identical study on Gram-positive bacteria, i.e., *S aureus* and *E. faecalis*, but the flavonoids they tested did not exhibit any lethal activity. Oksus et al. [71] determined the MIC of apigenin (54–219 µg/mL) against the *P. vulgaris*, *Ps. aeruginosa*, *E. coli*, and *K. pneumoniae* bacteria. Waage and Hedin [72] tested quercetin glycosides and noted that quercetin-3-O-rhamnoside exhibited the highest activity against *Ps. maltophilia* and *E. cloacae*. Liu et al. [73] observed that kaempferol glycosides were the most effective against Gram-positive bacteria. The MICs of the kaempferol and chloramphenicol glycosides against *B. cereus* were 16 µg/mL and 2 µg/mL, respectively [73]. The MICs of kaempferol and chloramphenicol glycosides against Gram-positive *S. aureus* bacteria were 32 μg/mL and 64 μg/mL, respectively [74]. Wang et al. [75] studied various methyl and acetyl derivatives of flavonoids and concluded that the presence of hydroxyl groups at the C-5 and C-7 positions was very important for the antimicrobial activity. The presence of an additional methoxy group at C7 or dihydroxy groups at C-3′ and C-4′ significantly reduced the antimicrobial activity [75]. Van Puyvelde et al. [76] noted the antibacterial activity of flavones, flavonols, flavanones, and isoflavones against *P. vulgaris* and *S. aureus*.

The canonical correlation analysis (CCA) showed a negative correlation between flavonoids (excluding luteolin and rutin), hydroxycinnamic acids (excluding caffeic acid), hydroxybenzoic acids (excluding chlorogenic and 4-hydroxybenzoic acids) and the presence of pathogenic and food spoilage microorganisms (Figure 2). The compounds with the greatest potential to inhibit microbial growth were kempferol, apigenin, protocatechuic, and ferulic acids.

There was an inversely proportional relation between glucose and citric and fumaric acids as well as between anthocyanins and malic and shikimic acids.

The canonical variate analysis (CVA) enabled the creation of a CCA model. Progressive stepwise analysis was used to determine which variables determined the activity of the pathogenic and food spoilage microorganisms to the greatest extent. All the variables were assessed and the ones that contributed most to the group discrimination based on the p and F values for the variable under analysis were included in the model. This process was repeated until the p value increased above 0.05 or the F value dropped below 2.00 for the variable under analysis.

The analysis determined homogeneous groups of elderberry extracts in terms of their content of compounds. There were three homogeneous groups in terms of the content of polyphenols and flavonoids: the first one was closely correlated with luteolin and vanilin–all the samples came from the Warmian-Masurian Voivodeship (Group I), the second one was correlated with chlorogenic, 4-hydroxybenzoic, and benzoic acids–the samples collected from the Lublin Voivodeship (Group II), the third one was correlated with the content of catechin, ferulic acid, and naringin (Kuyavian-Pomeranian Voivodeship, Łódź Voivodeship, and Greater Poland Voivodeship, but the sites were located in central Poland) (Group III). Table 6 and Table 7 list the statistical parameters for the detrended correspondence analysis. The list of Voivodeships of the analyzed samples is presented in Table 8.

The DCA revealed four homogeneous groups in terms of the antioxidative activity and the content of organic acids, pigments, and sugars in the elderberry extracts. As was the case with the previous model, groups I, II, and III were created. The fourth group was not related to the location, but it was similar in terms of the content of carotenoids, fructose, and fumaric and shikimic acids. In this case the location did not determine the distribution of compounds in the elderberries. Further research is necessary to identify factors influencing the degradation of bioactive compounds in elderberries.

In order to use elderberry extracts for health purposes, we should take into account the bioavailability of the active ingredients present in these extracts. Bioavailability can be defined as the amount or fraction of a compound that is released into the gastrointestinal tract and becomes available for absorption [77]. The research showed that the content of phenolic acids and flavonoids in the extracts determined their antibacterial activity. Scientific literature showed that phenolic acid can be absorbed in the stomach, small intestine, or both. [78]. The metabolization of flavonoids begins with hydrolysis of the glycosidic linkage, action that takes place in the intestine lumen or in enterocytes and then absorbed. These compounds, depending on their chemical structure, are hydrolyzed by endogenous human cytosolic β-glucosidase or by rhamnosidase, an enzyme produced by the intestinal microflora [79].

## 3. Materials and Methods

### 3.1. Research Material

The research was conducted on 500 g of wild elderberry fruits (*Sambucus nigra* L.) harvested in Poland in August and September 2019. The fruits were harvested in 36 places of their natural occurrence, which were marked with numbers in further part of this article. The fruits were harvested when they were fully ripe. After the harvest they were frozen and freeze-dried (temperature −53 °C, pressure 0.025 mbar). After drying the fruits were subjected to the extraction process.

The extracts were obtained with an innovative method of double hydrolysis (acidic and alkaline) and extraction of bioactive compounds, which were released from glycosidic bonds with diethyl ether. After evaporation the dry extracts were dissolved in deionized water and used in microbiological tests.

The extracts were tested for the minimum inhibitory concentration (MIC) against 7 bacterial strains and analyzed chemically to determine the content of phenolic compounds, pigments and antioxidative properties in a test with the ABTS^+^ radical.

### 3.2. Extraction Procedure

The phenolic compounds in the samples were analyzed after alkaline and acid hydrolysis. 10 g of the dehydrated plant material was placed in a 750 cm^3^ round bottom flask. The flask was placed under a reflux condenser. First, an alkaline hydrolysis was performed, followed by an acid hydrolysis. Next, 50 mL of distilled water and 200 mL of 2 M aqueous sodium hydroxide solution were added to the test tubes for the alkaline hydrolysis. The sealed flasks were heated in a heating mantle at 95 °C for 30 min. After cooling (approx. 20 min), the samples were neutralized with 100 mL of a 6 M aqueous hydrochloric acid solution (pH = 2). The samples were then cooled in ice water. Flavonoids were extracted from the inorganic (aqueous) phase with diethyl ether (2 × 100 mL). The ether extracts formed were continuously transferred to 250 cm 3 round bottom distillation flasks and evaporated on a rotary evaporator. Then acid hydrolysis was performed. For this, the aqueous phase was supplemented with 150 mL of 6 M aqueous hydrochloric acid solution and heated again at 95 °C for 30 min. After cooling with ice water, the samples were extracted with diethyl ether (2 × 100 mL). The produced ether extracts were continuously transferred to 250 cm^3^ round bottom distillation flasks and evaporated on a rotary evaporator. The extract was quantitatively transferred from the distillation flasks to 8 cm^3^ vials, washing the flasks 2 × 4 cm^3^ with diethyl ether. Then the extracts were dried under a stream of nitrogen at room temperature. The dry extracts were weighed and stored at −80 °C until further analysis (Table 8). Prior to subsequent analysis steps, the extracts were thawed under refrigeration for 12 h and then digested with a selected amount of MiliQ-grade distilled water to obtain the selected concentration. Elderberry extracts were digested to a volume of 25 cm^3^ in a volumetric flask.

#### Chromatographic Analysis

The analysis was performed using an Acquity H class UPLC system equipped with a Waters Acquity PDA detector (Milford, MA, USA). The chromatographic separation was performed on an Acquity UPLC^®^ BEH C18 column (Watersy, Dublin, Ireland) (100 mm × 2.1 mm, particle size 1.7 µm) (Watersy, Dublin, Ireland). The elution was carried out in a gradient using the following mobile phase composition: A: acetonitrile with 0.1% formic acid, B: 1% aqueous formic acid mixture (pH = 2). The concentrations of phenolic compounds were determined with the use of an internal standard at wavelengths λ = 320 nm and 280 nm and finally given in mg/1 g of extract. Compounds were identified by comparing the retention time of the analyzed peak with the standard retention time and by adding a specific amount of standard to the analyzed samples and repeating the analysis. The detection level was 1 µg/g.

### 3.3. Total Phenolic Content (TPC)

The total phenolic content was measured with the Folin-Ciocalteu reagent [80,81], 2 mL of the Folin-Ciocalteau reagent was added to 1 mL of the aqueous extract. After 3 min the reaction environment was alkalised by adding 10 mL of a 10% sodium carbonate solution. After 30 min the solutions were filled up to 25 mL and their absorbance was measured at a wavelength of λ = 765 nm, using a Hitachi U-2900 spectrophotometer (Schaumburg, IL, USA). The results were calculated as the mean of triplicates, in mg phenolic compounds per gram of raw material expressed as gallic acid equivalent (GAE).

### 3.4. Total Chlorophyll Content

The plant material was thoroughly ground in a mortar containing 3 mL of ethanol, a pinch of sand and a pinch of CaCO_3_. The solution was quantified into labelled centrifuge tubes. The mortar and pestle were rinsed with another 2 mL of alcohol, which was poured into the same centrifuge tubes. The tubes with the chlorophyll alcohol solution were capped and kept in a dark place until centrifugation. The samples were centrifuged at 9000 rpm for 10 min at room temperature. Next, the supernatant was quantitatively transferred into new labelled centrifuge tubes. Then, 1.9 mL of ethyl alcohol and 0.5 mL of the sample under analysis were poured into spectrophotometer cuvettes. The contents were mixed, and the chlorophyll content was determined with a UV/VIS Excellence 6850 spectrophotometer at wavelengths of: 645 nm, 649 nm, 654 nm, and 665 nm. The apparatus was zeroed at the specified wavelength on 2 mL of ethanol. All the measurements were triplicated. The formula below was used to calculate the content of chlorophyll *a* and *b*:Chlorophyll (a + b) = [(25.1 × A654) × (V: (1000 × W))] × 4 [mg g^−1^ fresh weight]
where:

A 645-665—absorbance measured at a wavelength of 649–665 nm,

V—total volume of the extract (mL),

W—weight of the sample (g) [82].

### 3.5. Total Anthocyanin Content (TAC)

The total anthocyanin content was measured with the spectrophotometric method described by Giusti and Wrolstad [80]. The results of the measurements were expressed as cyanidin-3-glucoside (C3G). 2 g of the product was collected and homogenized for 3 min (20,000 rpm) with 100 cm^3^ of a mixture of methanol and 1.5-molar hydrochloric acid (85:15). The homogenate was centrifuged for 20 min at 4000 rpm. A clear liquid was collected for analysis. The other products were diluted with buffers so as to make a spectrophotometric measurement within an absorbance range of 0.3–0.8. Depending on the sample, the dilution factor (DF) ranged from 12.5 to 20. All the measurements were triplicated with the UV VIS Excellence 6850 spectrophotometer [83,84,85].

### 3.6. Total Carotenoid Content

Determination of the total carotenoids and β-carotene content was carried out in accordance with the Polish standard [86].

### 3.7. Sugar Content

The content of glucose and fructose was measured in diluted and purified samples by means of HPLC with refractometric detection, using prepared standard curves. The chromatographic analysis was conducted under the following conditions: Shimadzu (Milton Keynes, England) apparatus, LC-20AD pump, RID—10A detector, Rezex RCM-Monosaccharide Ca^2+^ 300 × 7.8 mm column, column temperature 80 °C, mobile phase: deionized water, flow 0.8 cm^3^/min.

### 3.8. MIC Measurement

The minimum concentrations of the elderberry fruit extracts inhibiting (MIC) the growth of *Escherichia coli* (PCM 2793), *Salmonella enteritidis* (PCM 2548), *Proteus mirabilis* (PCM 1361), *Pseudomonas fluorescens* (PCM 2123), *Pseudomonas fragii* (PCM 1856), *Listeria innocua* (DSM 20649), and *Micrococcus luteus* (PCM 525) were measured with a Bioscreen C automated growth reader (Oy Growth Curves Ab Ltd.,TURKU, Finland). Bacterial inoculants and a series of diluted hydrated extracts were placed on a plate and incubated for 72 h under adequate conditions for individual groups of bacteria. The apparatus measured the optical density (OD) in each cuvette. The first concentration without turbidity was considered the MIC value, i.e., the minimum concentration at which the microorganisms used in the study were inactivated during 72-h incubation.

### 3.9. Statistical Analysis

Statistical analyses and models were based on discriminant analysis. The analyses showed which of the variables – pigments, sugars, organic acids, or polyphenols, may affect the activity of pathogenic and food-spoilage microorganisms. The model was constructed using canonical variate analysis (CVA)—the canonical variation of Fisher’s linear discriminant analysis (LDA) [87]. Detrended correspondence analysis (DCA) was used as an indirect method to prepare a diagram, ordering the samples under analysis. The borderline significance level was determined with the Monte Carlo permutation test (number of permutations: 9.999). The Canoco for Windows package and the Microsoft Excel spreadsheet was the software (Canoco 5) (accessed 3 October 2019) used for all comparisons, calculations, and graphic elements. The following tools from Canoco for Windows were used: Canoco for Windows 4.5, CanoDraw for Windows, and WcanoIMP.

## 4. Conclusions

The research showed that the content of phenolic acids and flavonoids in the extracts determined their antibacterial activity. Of the compounds under analysis, apigenin, kaempferol and ferulic, protocatechuic, and p-coumarin acids had the greatest influence on the high antibacterial activity of elderberry extracts. The statistical analyses did not show any significant influence of the location on the profile of bioactive compounds in the extracts. The extraction method presented in the study made it possible to obtain preparations containing free compounds with antioxidative and antibacterial properties. This method can facilitate the standardization of wild elderberry preparations.

## Figures and Tables

**Figure 1 molecules-26-02910-f001:**
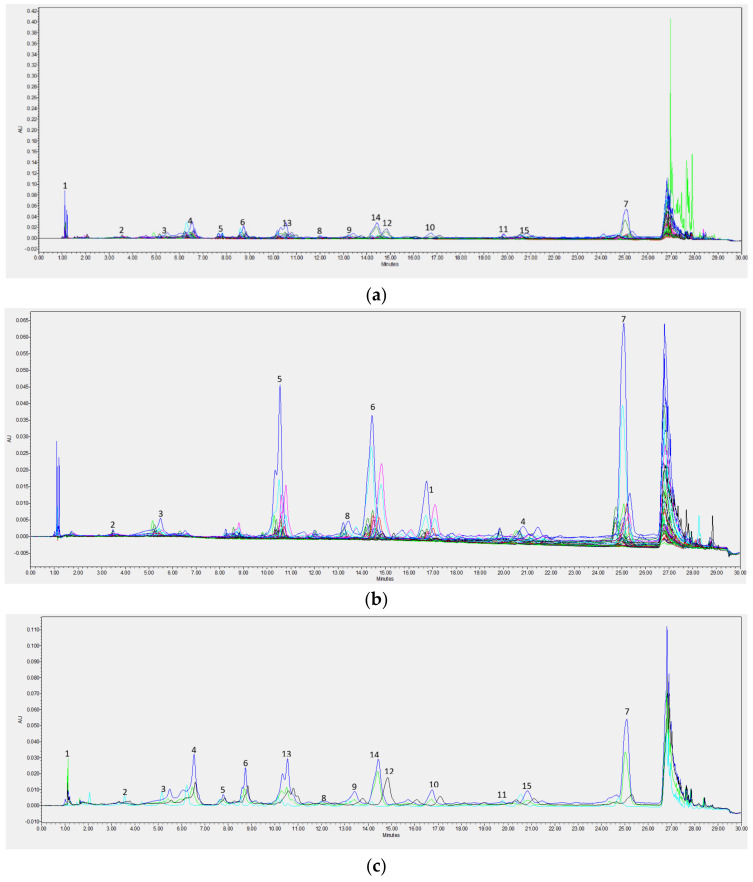
Representative chromatograms elderberry fruits extracts from UPLC/PAD of samples (**a**)–phenolic acids, (**b**)–flavonoids and a comparison of the most diversified samples (samples 5, 18, 20, and 36) (**c**)–phenolic acids (**d**)–flavonoids. (phenolic acids: 1–gallic acid, 2–4-hydroxybenzoic acid, 3–vanillic acid, 4–syringic acid, 5–vanilin, 6–benzoic acid, 7–chlorogenic acid, 8–protocatechuic acid, 9–salicylic acid, 10–caffeic acid, 11–p-coumaric acid, 12–ferulic acid, 13–sinapic acid, 14–t-cinnamic acid, 15–rosmarinic acid; flavonoids: 1–apigenin, 2–catechin, 3–kaempferol, 4–luteoline, 5–naringenin, 6–quercetin, 7–rutin, 8–vitexin).

**Figure 2 molecules-26-02910-f002:**
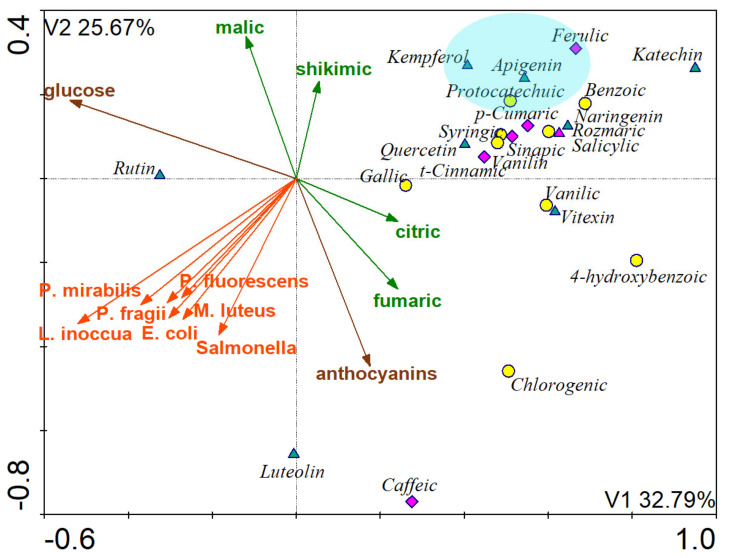
Canonical Correlation Analysis (CCA) (*n* = 38) Dependencies between groups of phenolic acids, organic acids, pigments, sugars and their influence on the development of pathogenic and food spoilage microorganisms.

**Figure 3 molecules-26-02910-f003:**
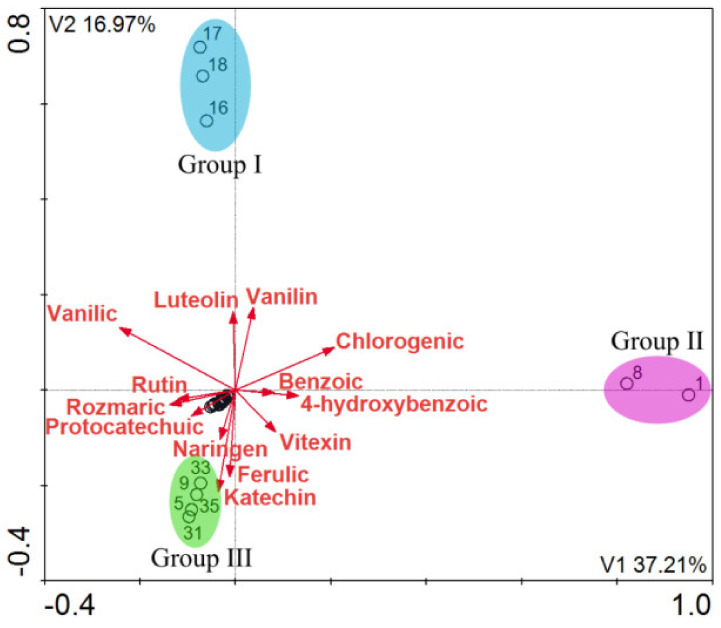
DCA (*n* = 38) A diagram showing the distribution of the elderberry extract samples in terms of the content of polyphenols and flavonoids.

**Figure 4 molecules-26-02910-f004:**
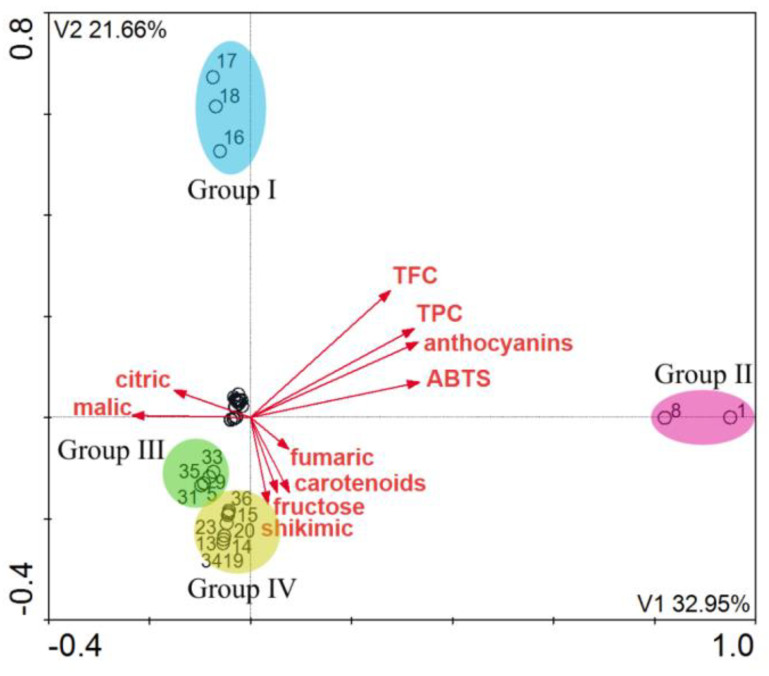
DCA (*n* = 38) A diagram showing the distribution of the elderberry extract samples in terms of the antioxidative activity and the content of organic acids, pigments, and sugars. Four homogeneous groups were identified.

**Table 1 molecules-26-02910-t001:** The total phenolic content (TPC), total phenolic acid content (TAC), total flavonoid content (TFC), and ABTS^+^ in the elderberry extracts.

	Min	Max	Mean	SD
ABTS^+^ (µmol Trolox equivalent/g extract)	647.21	721.25	**684.23**	1.09.1975
TPC (mg GAE/g extract)	3.23	18.90	**13.28**	4.34
TFC (mg RUTE/g extract)	11.25	263.50	**114.98**	64.14

**Table 2 molecules-26-02910-t002:** The content of phenolic acids in the elderberry extract.

[mg/g Extract]	Min	Max	Mean	SD
gallic	0.34	8.32	**3.34**	1.82
4-hydroxybenzoic	0.19	6.21	**1.36**	1.21
vanillic	0.02	0.25	**0.13**	0.06
syringic	0.42	3.09	**1.61**	0.75
vanillin	0.84	6.19	**3.13**	1.44
benzoic	2.09	30.96	**12.36**	5.98
chlorogenic	25.50	254.07	**139.09**	60.93
protocatechuic	0.24	1.61	**0.89**	0.38
salicylic	0.79	4.43	**2.56**	1.03
caffeic	0.26	5.53	**2.03**	1.40
p-cumaric	0.16	1.21	**0.61**	0.26
ferulic	11.20	67.09	**34.63**	14.84
sinapic	18.45	164.75	**72.84**	30.96
t-cinnamic	12.60	90.24	**51.29**	21.98
rosmarinic	0.16	2.24	**0.79**	0.41
**The content of flavonoids in the elderberry extract**				
**[mg/g extract]**	**Min**	**Max**	**Mean**	**SD**
apigenin	18.47	129.43	**66.74**	27.01
catechin	0.03	0.71	**0.23**	0.17
kaempferol	0.28	2.50	**1.01**	0.61
luteolin	2.76	49.13	**15.85**	10.24
naringenin	8.53	65.27	**30.71**	12.83
quercetin	97.38	504.88	**306.60**	125.02
rutin	124.98	2773.71	**1105.39**	604.12
vitexin	0.13	2.98	**1.35**	0.90

**Table 3 molecules-26-02910-t003:** The content of organic acids (LMWOAS) and sugars in the elderberry extract.

[mg/g Extract]	Min	Max	Mean	SD
citric	0.76	1.24	**1.03**	0.14
malic	0.23	0.37	**0.29**	0.05
shikimic	0.03	0.25	**0.14**	0.06
fumaric	0.03	0.12	**0.07**	0.03
glucose	3.08	7.40	**4.89**	1.44
fructose	3.88	9.38	**5.91**	1.77

**Table 4 molecules-26-02910-t004:** The content of pigments, carotenoids, chlorophyll, and anthocyanins in the elderberry extract.

[mg/g Extract]	Min	Max	Mean	SD
Total carotenoids	25.50	75.00	**47.93**	15.92
Total chlorophyll	0.08	0.54	**0.29**	0.14
TAC	75.18	149.78	**109.81**	22.62

**Table 5 molecules-26-02910-t005:** The minimum inhibitory concentrations (MIC) [%] of the elderberry extracts.

Tested Bacteria		Min	Max	Mean Value
Pathogenic bacteria	*E. coli* (PCM 2793)	0.05	0.5	**0.275**
	*S. enteritidis* (PCM 2548)	0.1	0.5	**0.3**
	*L. inoccua* (DSM 20649)	0.1	0.5	**0.3**
Food-spoilage bacteria	*P. fluorescens* (PCM 2123)	0.1	0.5	**0.3**
	*P. fragii* (PCM 1856)	0.05	0.5	**0.275**
	*P. mirabilis* (PCM 1361)	0.05	0.5	**0.275**
Control bacteria	*M. luteus* (PCM 525)	0.05	0.5	**0.275**

**Table 6 molecules-26-02910-t006:** Variance explained by the first four DCA axes (Figure 3).

Axes	1	2	3	4	Total Eigenvalues
Eigenvalue	0.3279	0.2567	0.0821	0.0458	2.115
Length of gradient	2.896	2.156	1.391	1.097	
Polyphenols and flavonoids– samples correlations	0.919	0.907	0.544	0.655	
Polyphenols and flavonoids– samples relation	35.7	37.1	0.0	0.0	

**Table 7 molecules-26-02910-t007:** Variance explained by the first four DCA axes (Figure 4).

Axes	1	2	3	4	Total Eigenvalues
Eigenvalue	0.3721	0.1697	0.6512	0.0329	2.059
Length of gradient	3.088	1.873	1.255	1.066	
Polyphenols and flavonoids–correlations in samples	0.951	0.742	0.421	0.235	
Polyphenols and flavonoids–correlations in samples	41.2	25.8	0.0	0.0	

**Table 8 molecules-26-02910-t008:** The mass of the extract obtained from individual *Sambucus nigra L.* samples.

No.	Location	Voivodeship	Mass of Extract [g]
1	51°24′99′′N 21°58′16′′E	Lublin Voivodeship	0.25
2	52°01′93′′N 17°78′44′′E	Greater Poland Voivodeship	0.19
3	52°65′67′′N 16°95′29′′E	Greater Poland Voivodeship	0.65
4	53°31′56′′N 20°67′35′′E	Greater Poland Voivodeship	0.39
5	52°99′64′′N 18°70′72′′E	Kuyavian-Pomeranian Voivodeship	0.20
6	49°39′96′′N 22°44′98′′E	Podkarpackie Voivodeship	0.79
7	49°27′54′′N 19°86′88′′E	Lesser Poland Voivodeship	0.48
8	51°25′05′′N 22°57′01′′E	Lublin Voivodeship	0.10
9	51°62′26′′N 17°94′28′′E	Greater Poland Voivodeship	0.69
10	50°29′68′′N 16°65′20′′E	Lower Silesian Voivodeship	0.44
11	53°92′82′′N 14°44′89′′E	West Pomeranian Voivodeship	0.80
12	53°91′31′′N 14°52′00′′E	West Pomeranian Voivodeship	1.05
13	53°47′30′′N 17°89′64′′E	Kuyavian-Pomeranian Voivodeship	1.83
14	53°48′46′′N 18°07′17′′E	Kuyavian-Pomeranian Voivodeship	0.74
15	53°77′66′′N 20°47′65′′E	Warmian-Masurian Voivodeship	0.61
16	53°39′84′′N 20°94′62′′E	Warmian-Masurian Voivodeship	0.16
17	53°58′34′′N 20°28′16′′E	Warmian-Masurian Voivodeship	1.04
18	54°21′38′′N 21°74′16′′E	Warmian-Masurian Voivodeship	0.28
19	53°81′29′′N 20°35′80′′E	Warmian-Masurian Voivodeship	0.40
20	53°59′70′′N 19°85′43′′E	Warmian-Masurian Voivodeship	1.08
21	54°47′25′′N 16°63′07′′E	West Pomeranian Voivodeship	0.12
22	51°30′05′′N 16°83′01′′E	Lower Silesian Voivodeship	0.37
23	54°16′88′′N 17°49′22′′E	Pomeranian Voivodeship	0.50
24	53°26′97′′N 16°46′70′′E	West Pomeranian Voivodeship	0.16
25	52°39′91′′N 16°71′89′′E	Greater Poland Voivodeship	1.38
26	52°97′29′′N 16°54′44′′E	Greater Poland Voivodeship	0.18
27	53°27′61′′N 15°46′33′′E	West Pomeranian Voivodeship	0.71
28	52°77′12′′N 16°87′97′′E	Greater Poland Voivodeship	0.94
29	52°80′71′′N 17°19′73′′E	Greater Poland Voivodeship	0.11
30	51°76′81′′N 15°87′48′′E	Lubusz Voivodeship	1.05
31	52°10′76′′N 19°94′47′′E	Łódź Voivodeship	0.56
32	53°24′68′′N 17°01′70′′E	Greater Poland Voivodeship	0.22
33	52°04′58′′N 18°36′86′′E	Greater Poland Voivodeship	0.57
34	52°53′95′′N 16°26′42′′E	Greater Poland Voivodeship	0.16
35	51°76′87′′N 19°45′69′′E	Łódź Voivodeship	0.52
36	52°47′75′′N 16°87′72′′E	Greater Poland Voivodeship	2.12

## Data Availability

Not applicable.

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
