# Peer review of "Sambucus Nigra Extracts–Natural Antioxidants and Antimicrobial Compounds"

_molecules, 2021, doi:10.3390/molecules26102910_

Round 1

Reviewer 1 Report

The manuscript describes a study on the evaluation of the antioxidant and antimicrobial activity of elderberry extracts from different regions in Poland. The properties are correlated to the chemical composition,  and more specifically to the content of phenolic acids and flavonoids.

Indeed, as reported by various contributions, it is known a strict relationship between these natural organic compounds and their properties (as mentioned by the authors).

The analyses on the extracts against selected bacteria are rather accurate, and in line with the conclusions presented by the authors. Furthermore, the location has no significant correlation with the bioactivity of the analysed extracts.

I may recommend the publication in Molecules after minor revision (see below).

-           Pag. 4, Figure 1: the type of Chromatograms should be specified.

-           Pag. 9, Section 3.2. Extraction procedure: since the authors report a novel method to obtain the extracts (double hydrolysis with NaOH and HCl), I suggest to briefly describe what is obtained after first treatment with sodium hydroxide: for example, a general structure of the main components formed after alkaline treatment could be shown (carboxylates, phenoxides), depending on their pKa.

In the same page (bottom) is reported “…were extracted from the inorganic phase…”: it means “water phase”? Please specify.

-           Page 10: it is not very clear which is the aqueous phase treated with HCl (6M):  please specify.

Author Response

Author's Reply to the Review Report (Reviewer 1)

Dear Reviewer

Thank You for positive opinion and all comments.

We amended the relevant parts in the manuscript according to your advice. The detailed response is as follows:

  • Pag. 4, Figure 1: the type of Chromatograms should be specified.

Chromatograms UPLC/PDA

  • Pag. 9, Section 3.2. Extraction procedure: since the authors report a novel method to obtain the extracts (double hydrolysis with NaOH and HCl), I suggest to briefly describe what is obtained after first treatment with sodium hydroxide: for example, a general structure of the main components formed after alkaline treatment could be shown (carboxylates, phenoxides), depending on their pKa.

Amendments were made to the manuscript in section 3.2 Extraction procedure

  • In the same page (bottom) is reported “…were extracted from the inorganic phase…”: it means “water phase”? Please specify.

The inorganic phase means water phase

Reviewer 2 Report

This manuscript describes natural antioxidants and antimicrobial compounds in extracts from Sambucus nigra collected from 36 locations in Poland. This manuscript reviews more than 84 articles and provide a complex survey of current literature dealing with this topic. The topic of this manuscript is up to date, interesting and well suited for the journal Molecules. The manuscript is well written and divided into 4 parts.  The text is clear and easy to read. For better understanding authors used 4 illustrations and 8 tables. This aid the readers understanding. I suggest checking for some small spelling mistakes and grammar errors. Otherwise, I have no major concerns about this manuscript and I recommend it for publication.

Author Response

Author's Reply to the Review Report (Reviewer 2)

Dear Reviewer

Thank You for positive opinion and all comments.

We checked and amended some small spelling mistakes and grammar errors the relevant parts in the manuscript according to your advice.

Reviewer 3 Report

Dear Authors,

After the review process, I have several comments: you should include numerical data in the abstract; the aim of the paper is relevant to another research direction - microbiota modulation. In order to increase the study valorization, you should include comments about correlations between microbiota bioactivity and bioavailability of functional compounds and the bioactive potential of functional products, and the bioavailability of phenolic compounds.

Best regards,

Author Response

Author's Reply to the Review Report (Reviewer 3)

Dear Reviewer

Thank You for positive opinion and all comments.

Numerical data have been added in the abstract (page 1 text in a different color). The major weaknesses which reviewer encountered in the manuscript were corrected (page 10 text in a different color).

.

Round 2

Reviewer 3 Report

No other comments.

Author Response

Thank you for checking the manuscript. The introduction has been supplemented as suggested by the reviewer (page 2 in manuscript) 

Best regards,

LIdia Szwajkowska-Michałek

Department of Chemistry,

Faculty of Forestry and Wood Technology,

Poznan University of Life Sciences

ul. Wojska Polskiego 75, Poznań Poland